# Shift in Epidemiology of Cryptococcal Infections in Ottawa with High Mortality in Non-HIV Immunocompromised Patients

**DOI:** 10.3390/jof5040104

**Published:** 2019-11-10

**Authors:** Vishesh Patel, Marc Desjardins, Juthaporn Cowan

**Affiliations:** 1Faculty of Medicine, University of Ottawa, Ottawa, ON K1H 8L6, Canada; vpate093@uottawa.ca; 2Eastern Ontario Regional Laboratory Association, Department of Pathology and Laboratory Medicine, University of Ottawa, Ottawa, ON K1H 8L6, Canada; madesjardins@eorla.ca; 3Division of Infectious Diseases, Department of Medicine, University of Ottawa, Ottawa, ON K1H 8L6, Canada; 4Department of Biochemistry, Microbiology and Immunology, University of Ottawa, Ottawa, ON K1H 8L1, Canada; 5Clinical Epidemiology Program, Ottawa Hospital Research Institute, Ottawa, ON K1H 8L6, Canada

**Keywords:** cryptococcosis, human immunodeficiency virus, hematological malignancy, solid organ transplantation, epidemiology, immunocompromised, mortality

## Abstract

*Cryptococcus neoformans* is a fungus that can cause life-threatening infections. While human immunodeficiency virus (HIV)-positive status historically had the highest risk for cryptococcal infection and was associated with high mortality rates, there have been changes in HIV treatment and the epidemiology of other acquired immunodeficiencies, such as hematological malignancies. We conducted a retrospective case series analysis of patients who had cryptococcal infections documented at the Ottawa Hospital from 2005 to 2017. The Ottawa Hospital is a tertiary care hospital and provides complex care such as chemotherapy and transplantations. There were 28 confirmed cryptococcal infections. The most common underlying condition associated with cryptococcal infection was hematological malignancy (*n* = 8.29%), followed by HIV (*n* = 5.18%) and solid organ transplantation (*n* = 4.14%). Furthermore, while there was a decrease in the number of cryptococcal infections in HIV patients after 2010 (four to one case), the number of cases in non-HIV immunocompromised patients increased from four in the years 2005–2010 to fourteen in 2011–2017. There were nine cryptococcal-attributable deaths. The case fatality rate was highest among patients with underlying hematological malignancies (63%), followed by solid organ transplant (50%) and HIV patients (20%). In conclusion, this study showed that there may be an epidemiological shift of cryptococcal infection in Ottawa. Additionally, infections may be associated with a worse prognosis in patients with a hematological malignancy and solid organ transplant than in patients with HIV infection in the modern era. Better prevention and/or treatment is warranted for high-risk populations.

## 1. Introduction

Cryptococcosis is an infection that is caused by the fungal species *Cryptococcus neoformans* or *Cryptococcus gattii.* Between the two, *C. neoformans* is the more common pathogen with a worldwide distribution and a tendency to infect immunocompromised hosts [1,2]. Historically, infection rates were highest amongst human immunodeficiency virus (HIV)-positive patients and *C. neoformans* was the leading cause of fungal-related death in these patients due to cryptococcal meningitis and sepsis [3,4]. However, improvements such as highly active antiretroviral therapy (HAART) have reduced HIV-related cryptococcal mortality in nations where the highest standard of care is widely available compared to developing countries [5]. Other than HIV infection, several immunocompromised conditions have been shown to be risk factors for cryptococcal infections. For example, solid organ transplants, hematological malignancies, autoimmune diseases, and many of the treatments associated with these conditions can lead to reduced immune function and increased risk of cryptococcosis [6]. 

Several studies reported on changes in epidemiology and outcomes of cryptococcal infection in non-HIV populations during different time periods. For example, diabetes was found to be the most common underlying condition associated with cryptococcal infection in non-HIV patients in the Atlanta and Houston metropolitan areas between 1992−2000 [7], while others found solid organ transplantation and hematological malignancies to be the most common underlying conditions in the highly active antiretroviral and advanced cancer therapy era [6,8]. In addition, hematological malignancy was identified as a predictor of mortality in the late 1990s [9], but this association was not found in the 2000s by the same research group [10]. There has not been any report on this epidemiological shift of cryptococcal infection in Canada. Based on the effectiveness of antiretroviral treatment and improvements in healthcare for HIV-infected patients, we hypothesized that cryptococcal infection was more prevalent in the non-HIV than in the HIV population. Furthermore, we attempted to compare mortality among patients with various immunocompromised states. 

## 2. Materials and Methods

### 2.1. Study Design

This study was conducted using a hospital-based retrospective case series analysis of patients who had cryptococcal infections. Records from The Ottawa Hospital Data Warehouse were screened to identify patients who had a suspected cryptococcal infection in their medical records between the time period of January 2005 to December 2017. The Ottawa Hospital is a tertiary care hospital and a referral center within Ottawa for Eastern Ontario and parts of Northern Ontario. Complex care and treatments such as chemotherapy and transplantation in this region occur at the Ottawa Hospital. The initial screen of positive cultures or cryptococcal antigen detection in any clinical samples resulted in a list of 38 potential patients. From this list, the medical records were reviewed individually by the corresponding author to distinguish *C. neoformans* infections from patients who did not have cryptococcosis. *C. neoformans* isolated from blood, cerebrospinal fluid (CSF), or tissue biopsy samples were always considered as a true infection. It is known that clinical manifestations of pulmonary cryptococcosis range from asymptomatic colonization of the airways to life-threatening infections [1,2]. Therefore, *C. neoformans* isolated from respiratory tract sample was considered to be probable if the patient was immunosuppressed without history of bronchiectasis, or the treating physicians felt that patient’s symptoms were explained by cryptococcal infection and decided to treat as such. Of 38, ten had an alternative diagnosis and did well without definitive antifungal treatment for cryptococcus; therefore, they were excluded from our analysis. All but two samples of the ten excluded cases were collected from the respiratory tract. The remaining 28 patients were identified as having cryptococcal infections within our time period. Each patient had documented *C. neoformans* growth from cultures of either blood, bronchoalveolar lavage, or cerebrospinal fluid. The date of infection was defined as the first date that clinical disease was confirmed with positive laboratory cultures of *C. neoformans*. A thorough review of each patient’s relevant medical records was conducted, including laboratory results and physician notes, to gather the data used in this study. Reports regarding patient outcomes were recorded along with the comorbidities that they had at the time of cryptococcal infection. The clinical outcome was classified as either a resolution of the infection which was documented by an official physician report or a treatment failure leading to mortality as documented by a death certificate.

### 2.2. Data Analysis

Descriptive analysis was conducted for each variable. All data was reported using both the raw number of patients in that group as well as a percentage of the total number of patients. Follow up time was calculated as the time between the first reported note of cryptococcal infection to the date of death or last date of available medical record for patients who survived the cryptococcosis. Resolution of infection was determined based on physician reports of successful treatment and discontinuation of antifungal medication. 

### 2.3. Ethical Approval

This study was reviewed and approved by the hospital’s research ethics board. The protocol identification number was 20170859-01H. Informed consent was waived due to the nature of the retrospective case series study.

## 3. Results

Each of the 28 patients with cryptococcosis were divided into groups based on the comorbidity and associated treatments that were believed to be the risk factors for infection (Figure 1). Five (18%) of the patients were determined to be HIV positive at the time of infection. CD4 cell count data was available in four of five patients with a mean and median of 95 and 41 cells/μL respectively. Other immunocompromised conditions were hematological malignancy (*n* = 8.29%), solid organ transplant (*n* = 4.14%), autoimmune (*n* = 3.11%), and solid tumors (*n* = 2.7%). The remaining six patients had other comorbidities not typically classified as immunocompromised conditions including chronic lung disease (*n* = 4.14%) and end-stage renal disease (*n* = 2.7%). From these six, five had confirmed negative HIV serology while one was never tested for HIV. Data on age, sex, site of cryptococcosis, underlying medical disease, and treatment for the underlying condition for each patient is shown in Table 1. 

Each case of cryptococcosis was categorized based on the date of infection and the associated risk factor (Figure 2). The majority of infections in HIV-positive patients were seen before 2010 with only one case from 2011 to 2017. In comparison, there were two cases associated with hematological malignancy before 2010 and six cases from 2011 to 2017. Collectively, the solid organ transplant, autoimmune, and solid tumor groups had a total of two cases from 2005 to 2010 while the remaining seven patients were infected from 2011 to 2017. There was an average of 49,291 and 52,056.6 admissions/year to non-surgical units during 2005–2010 and 2011–2017, respectively. 

At the time of conducting this retrospective study, 12 out of the 28 patients were deceased with a mean and median follow-up time of 54 and 32.5 months (range 1 day to 172 months), respectively. None of the patients had a recurrence of cryptococossis during the follow-up period. A total of nine patients died due to cryptococcosis while the remaining three died due to other diseases that were independent of the infection. Of the nine cryptococcal related deaths, only one was an HIV-infected patient, while five had hematological malignancy and two were solid organ transplant recipients. The case fatality rate (Figure 3) was highest in patients with hematological malignancy (63%), followed by solid organ transplantation (50%) and HIV (20%). The HIV patient who died was not known to be HIV positive at the time of presentation with severe disseminated cryptococcosis and subsequently died within a few days. The number of deaths in the HIV group was comparable to the deaths in the group with cryptococcal infections that were not associated with typical comorbidities. No deaths were found in patients with autoimmune diseases or solid tumors.

## 4. Discussion

In this study, we reviewed the recent cases of cryptococcosis in our academic center to determine if there had been a change in the epidemiology of cryptococcal infections. Based on clinical experience, we hypothesized that cryptococcal infections associated with non-HIV related immunological deficits, particularly hematological malignancies, may have become more common. This hypothesis was supported by our data as a greater proportion of the cryptococcal infections were associated with hematological malignancies compared to HIV infections. HIV-positive status only accounted for 18% of cryptococcal infection in our study. Additionally, the majority of cryptococcosis cases in HIV-positive individuals occurred before 2010 while the majority of cases associated with non-HIV risk factors were reported after 2010. This distribution of cases suggests that while being HIV positive may have been an important risk factor in the past, it may not be the primary concern going forward given available early detection and effective treatment. Instead, awareness should be raised regarding other risk factors such as hematological malignancies, organ transplants, and their respective treatments. Our data was not biased by changes in the number of beds or number of admissions in the hematology unit, as the number of admissions to non-surgical units (including hematology unit) and to the hematology ward only at our hospital were proportionally similar. There were an increased number of admissions by 5807 and 398 patients/year to non-surgical and to hematology units, respectively, from 2005 to 2017. We do not believe that there was an increase in non-HIV medical conditions per se, but we believe that these patients are sicker and more immunosuppressed than those we saw a decade ago. 

Furthermore, this study also shows that cryptococcal infections may be associated with worse outcomes in patients with hematological malignancy compared to infections in other groups such as HIV-positive individuals. With 63% of the patients with hematological malignancy unable to recover from their cryptococcosis, this secondary immunocompromised state may be severely deficient in protection against *C. neoformans*. Additionally, the hematological malignancies reported in this study were treated with various chemotherapeutic agents. These medications can interfere with the normal immune response to pathogens such as *C. neoformans* and further increase the risk of infection. The use of treatments known to reduce immune function was also reported in patients with solid organ transplants and autoimmune conditions. Although the treatments were not separated from the associated conditions during this study, they may contribute to a substantial increase in the overall risk of cryptococcosis in non-HIV patients. Chronic lymphocytic leukemia (CLL) is a disease of B-lymphocytes and is generally considered to be a disease of humoral immune impairment; however, it has been shown that untreated CLL patients have lower percentage of CD4+ T-cells than in healthy controls [11]. The quality of T-cells was also impaired with decreased proliferation and cytokine production. It has also been proposed that dysfunctional T-cells are caused by impaired synapse between T-cells and antigen-presenting cells [12]. Therefore, CLL patients can have cellular mediated immunity impairment in addition to humoral immunity impairment. We reported previously of the three untreated CLL cases that developed cryptococcal infection [13]. Among hematological cancer patients in our cohort, almost a third of patients were not treated with some kind of chemotherapy in the last six months prior to cryptococcal infection. It is possible that cryptococcal infection can occur in this patient population much more often than our stereotypic idea of infection in only chemotherapy-treated patients. However, we cannot exclude the long-term effect of chemotherapy with our limited data. It is important to note that all nine deaths related to cryptococcal infection occurred rapidly within three months regardless of underlying medical conditions. 

Data from a single center is a limitation for generalizability. However, the Ottawa Hospital is a referral center for the region and provides a complex care as well as transplantation (hematopoietic stem cell, kidney and heart). The patient demographic is similar to other tertiary hospitals in Canada. In addition, a similar study found that the odds ratio for mortality was higher in non-HIV and non-organ transplant patients (NHNT) than it was in HIV-positive patients or organ transplant patients [10]. Although the authors did not report the odds ratio for hematological malignancy specifically, their data complements our study by the fact that HIV-associated cryptococcal infection does not result in the highest mortality. Our data also suggests that although the occurrence of cryptococcal infection in solid organ transplant was comparable to HIV patients, the mortality rate was much higher in solid organ transplant (50% vs. 20%). Another recent study also found non-HIV-related cryptococcal infections to have a higher 30-day mortality than HIV patients and this was associated with a prolonged time to diagnosis in the non-HIV group [14]. Interestingly, hematological malignancy was not found to be associated with mortality in another study [10]. As reported by the researchers, this was likely due to the utilization of oral triazole antifungal prophylaxis in hematological malignancy patients at that academic center. 

We did not find any cryptococcal infection in hematopoietic stem cell transplant recipients. This is likely because fluconazole is part of the antimicrobial prophylactic protocol for all transplant recipients at our center. This suggests that azole agents may be used to prevent cryptococcal infection in patients with hematological malignancies. However, the results of this study are limited because it was restricted to one academic center. Further research is needed to identify high-risk populations who may benefit from antifungal prophylaxis as well as other diagnostic and treatment options to improve clinical outcomes.

## 5. Conclusions

While cryptococcal infections seem to be less common in HIV patients compared to the pre-HAART era, it may be important to identify other risk factors for cryptococcosis such that infections can be diagnosed earlier and complications can be limited. Based on the findings of this study, hematological malignancies and their associated chemotherapeutic treatments have been identified as a risk factor that should be considered when making clinical decisions, as they may lead to a greater number of infections and deaths. Although further research is needed to determine efficacy, early use of antifungal treatment should be considered in these high-risk patients. 

## Figures and Tables

**Figure 1 jof-05-00104-f001:**
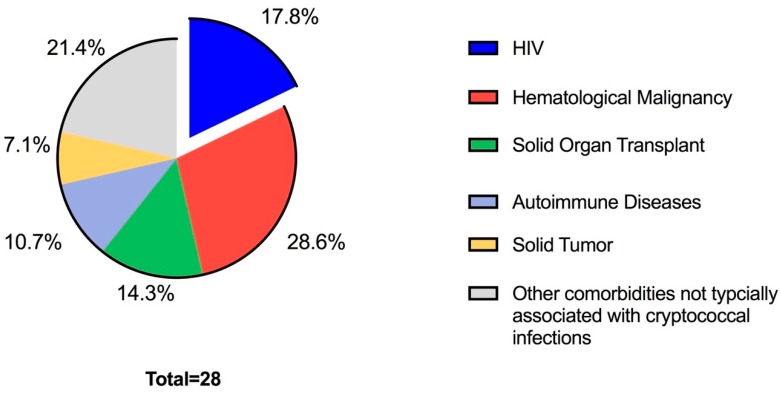
Percentage of the total number of cryptococcal infections represented by each underlying comorbidity.

**Figure 2 jof-05-00104-f002:**
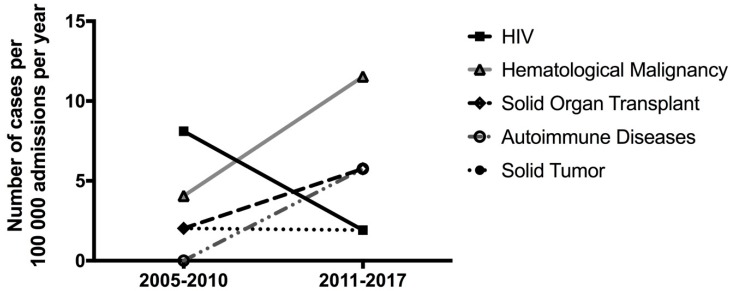
Number of cryptococcal infection cases per 100,000 admissions per year for each underlying comorbidity organized by the decade in which the case was reported.

**Figure 3 jof-05-00104-f003:**
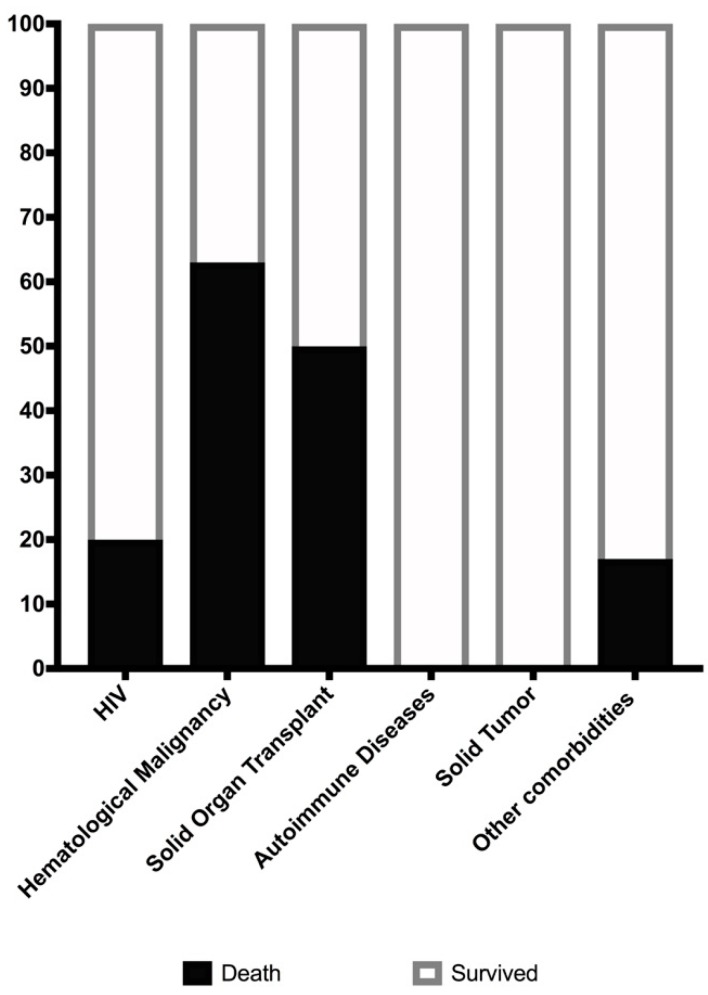
Cryptococcal infection case fatality rate for each examined comorbidity.

**Table 1 jof-05-00104-t001:** Patient characteristics and outcomes.

Age	Sex	Underlying Medical Disease	Treatment of Underlying Condition within 6 Months of Cryptococcal Diagnosis	Site of Positive Culture	Treatment of Cryptococcal Infection	Outcome Following Cryptococcal Infection	Follow-Up Time
56	M	HIV	Anti-retroviral therapy	CSF	Amphotericin B and fluconazole	Resolved	129 months
35	F	HIV	None	CSF	None	Death	2 days
53	M	HIV	None	CSF and blood	Fluconazole	Resolved	132 months
54	M	HIV	None	CSF	Amphotericin B and fluconazole	Resolved	98 months
59	M	HIV	Anti-retroviral therapy	Bronchial washing	Fluconazole	Resolved	142 months
72	M	Acute myeloid leukemia	Azacitidine	Blood	None	Death	9 days
77	M	Multiple myeloma	Melphalan and dexamethasone	Blood	None	Death	7 days
62	M	Hairy cell leukemia, and cirrhosis from chronic hepatitis B	None	Blood	Amphotericin B and fluconazole	Death	6 days
63	M	Chronic lymphocytic leukemia	None	Blood, CSF, Skin	Amphotericin B and fluconazole	Resolved	32 months
85	M	Chronic lymphocytic leukemia and bladder cancer	Prednisone 5 mg daily	BAL	Itraconazole then fluconazole	Resolved but deceased 2 years later from cancer	33 months
43	M	Acute myeloid leukemia	Cytarabine and idarubicin (1 cycle)	Sputum	Amphotericin B, and fluconazole	Resolved	33 months
84	M	Chronic lymphocytic leukemia	No documentation	Blood	None	Death	7 days
72	M	Chronic lymphocytic leukemia	Chlorambucil and prednisone	Blood	Amphotericin B	Death	18 days
64	M	Kidney transplantation	Prednisone, tacrolimus and mycophenolate	Blood	None	Death	1 day
61	F	Kidney transplantation	Prednisone, tacrolimus and mycophenolate	Blood	Fluconazole	Death	86 days
70	F	Liver and kidney transplantation	Prednisone, tacrolimus and mycophenolate	BAL	None*	Resolved	84 months
64	M	Kidney transplantation	Prednisone, tacrolimus and mycophenolate	CSF	Fluconazole	Resolved	172 months
53	M	Sarcoidosis	Prednisone	BAL	Fluconazole	Resolved	99 months
72	F	Sjogren’s syndrome and rheumatoid arthritis	Prednisone, methotrexate, and leflunomide	BAL	Fluconazole	Resolved	32 months
74	M	Sarcoidosis	Prednisone and azathioprine	Skin	Fluconazole	Resolved	25 months
63	M	Lung cancer	None	BAL	None**	Death from metastatic lung cancer	45 days
65	F	Papillary thyroid cancer	None	BAL	Fluconazole	Resolved	113 months
77	M	Asthma	Ciclesonide, Fluticasone, and Salbutamol	BAL	Fluconazole	Resolved	40 months
59	F	Asthma	Fluticasone and Salbutamol	BAL	Fluconazole	Resolved	131 months
73	F	End stage renal disease	Hemodialysis	Blood	None	Death	3 days
53	M	Diabetes	Insulin glargine and insulin aspart	BAL	Fluconazole	Resolved	62 months
73	M	End stage renal disease	Peritoneal dialysis	Peritoneal fluid	Fluconazole	Resolved	104 months
73	M	Chronic obstructive pulmonary disease, chronic kidney disease	Fluticasone, Salbutamol, and Tiotropium	BAL	Fluconazole	Resolved but deceased later from abdominal aortic aneurysm rupture	22 months

CSF = cerebrospinal fluid; BAL = bronchoalveolar lavage. * It appeared that the treating team might not be aware of the culture result. The patient was treated with valganciclovir for cytomegalovirus (CMV) viremia at that time as well. CMV viremia improved but his respiratory symptoms were slow to improve. It was unclear whether anti-rejection medications were reduced at the time of CMV viremia which might also have helped pulmonary infection. ** By the time the culture became available, the patient was palliative and receiving end-of-life care.

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
