# Peer review of "Shift in Epidemiology of Cryptococcal Infections in Ottawa with High Mortality in Non-HIV Immunocompromised Patients"

_jof, 2019, doi:10.3390/jof5040104_

Round 1

Reviewer 1 Report

This paper describes a shift in the epidemiology of cryptococcal infections in the Ottawa Hospital in Canada.  It is important in that it demonstrates that other risk factors for cryptococcal disease need to be considered besides HIV infection, now that the prevalence of cryptococcosis in HIV is decreasing.  The authors also point out the important fact, previously identified, that non-HIV patients with cryptococcal disease have poorer outcomes; awareness of this fact is important because delays in diagnosis may account for part of this. 

My major concern about his paper is that the authors implicate such things as hematological malignancies as "other risk factors" (see line 158) but totally ignore the role of the therapies for these disorders as the most important contributing factor.  So, for instance, 4 of the patients in Table 1 have chronic lymphocytic leukemia.  This disease, in and of itself, should not predispose to cryptococcosis as it is a humoral immunity disorder. However, many patients get fludarabine for therapy; this drug has profound effects on T-cells and T-cell immunity, a known risk for susceptibility to fungal diseases like cryptococcosis.  Similarly, for patients in the table with solid organ transplantation, the steroids, calcineurin inhibitors and other agents are the important risks.  Sarcoidosis is associated with CD4 lymphocyte depletion or steroid therapies, which would be the most important risks.  Therefore, the authors need to add another column to their table indicating what therapies the patients received for their underlying disorders.  Furthermore, they need to change the discussion to clarify the very important role of therapies for these disorders as the more important risks than the diseases themselves. If a CD4 count is available for the sarcoidosis patient, that should be provided.

Author Response

This paper describes a shift in the epidemiology of cryptococcal infections in the Ottawa Hospital in Canada.  It is important in that it demonstrates that other risk factors for cryptococcal disease need to be considered besides HIV infection, now that the prevalence of cryptococcosis in HIV is decreasing.  The authors also point out the important fact, previously identified, that non-HIV patients with cryptococcal disease have poorer outcomes; awareness of this fact is important because delays in diagnosis may account for part of this.

 My major concern about his paper is that the authors implicate such things as hematological malignancies as "other risk factors" (see line 158) but totally ignore the role of the therapies for these disorders as the most important contributing factor.  So, for instance, 4 of the patients in Table 1 have chronic lymphocytic leukemia.  This disease, in and of itself, should not predispose to cryptococcosis as it is a humoral immunity disorder. However, many patients get fludarabine for therapy; this drug has profound effects on T-cells and T-cell immunity, a known risk for susceptibility to fungal diseases like cryptococcosis.  Similarly, for patients in the table with solid organ transplantation, the steroids, calcineurin inhibitors and other agents are the important risks.  Sarcoidosis is associated with CD4 lymphocyte depletion or steroid therapies, which would be the most important risks.  Therefore, the authors need to add another column to their table indicating what therapies the patients received for their underlying disorders.  Furthermore, they need to change the discussion to clarify the very important role of therapies for these disorders as the more important risks than the diseases themselves. If a CD4 count is available for the sarcoidosis patient, that should be provided.

 Thank you for this important point. We have incorporated discussions around these issues in the discussion.

We have also included a column of treatment of underlying conditions in the Table 1 as suggested.

Unfortunately, we do not have records of CD4 cell count in patients with sarcoidosis. However, their lymphocyte counts are normal acknowledging that CD4 T cell is only a subset of lymphocytes.

Reviewer 2 Report

The manuscript submitted by Patel and colleagues reports the results of a retrospective case series of 28 patients with cryptococcal infections at Ottawa Hospital between 2005 and 2017. The most common underlying condition was hematological malignancy (n = 8, 29%), followed by HIV (n = 5, 18%) and solid organ transplantation (n = 4, 14%) with a notable increase in cases in non-HIV immunocompromised patients from four during 2005‐2010 to fourteen in 2011‐2017. There were nine attributable deaths, and the case fatality rate was highest among patients with hematological malignancies (63%), followed by solid organ transplant (50%) and HIV patients (20%).

Altogether, this is an interesting short report that emphasizes the risk of non-HIV infected, severely immunocompromised patients for cryptococcal diseases with dismal prognosis.

Abstract: May be focused to the essential information. Introduction: Appropriate in size and content. Methods: Please provide background information on the patient population and medical interventions available at your hospital. This is essential to understand the data. Methods/Results: I suggest to create a more informative and complete table detailing the basic information on each of the 28 cases (f.e., age, gender, underlying condition, immunosuppressive treatment, cryptococcal disease and diagnostic support, treatment, outcome). Figure 1: Pl. modify the legend for group 6 to ‘other comorbidities not typically associated with cryptococcal diseases’. Methods/Results: What were the criteria to consider a positive BAL finding as diagnostic of disease? Pl. clearly define the disease entities that you used for categorization in the methods section. Figure 2: A denominator is needed in order to be able to make any epidemiological assessment (f.e. hospital days, inpatient admissions). Results: What was the median (range) of follow-up of the cohort, and at what time point did you assess responses to therapy? Pl. clarify.

Author Response

The manuscript submitted by Patel and colleagues reports the results of a retrospective case series of 28 patients with cryptococcal infections at Ottawa Hospital between 2005 and 2017. The most common underlying condition was hematological malignancy (n = 8, 29%), followed by HIV (n = 5, 18%) and solid organ transplantation (n = 4, 14%) with a notable increase in cases in non-HIV immunocompromised patients from four during 2005‐2010 to fourteen in 2011‐2017. There were nine attributable deaths, and the case fatality rate was highest among patients with hematological malignancies (63%), followed by solid organ transplant (50%) and HIV patients (20%).

Altogether, this is an interesting short report that emphasizes the risk of non-HIV infected, severely immunocompromised patients for cryptococcal diseases with dismal prognosis.

Abstract: May be focused to the essential information. Introduction: Appropriate in size and content. Methods: Please provide background information on the patient population and medical interventions available at your hospital. This is essential to understand the data.

Thank you for your suggestion. We have included a brief description about our hospital in the methods section.

Methods/Results: I suggest to create a more informative and complete table detailing the basic information on each of the 28 cases (f.e., age, gender, underlying condition, immunosuppressive treatment, cryptococcal disease and diagnostic support, treatment, outcome).

Thank you for this input. Based on the feedback, we have added additional information to complete Table 1. This will give the readers more insight on the nature of each case and the outcomes.

Figure 1: Pl. modify the legend for group 6 to ‘other comorbidities not typically associated with cryptococcal diseases’.

The legend was changed as suggested.

Methods/Results: What were the criteria to consider a positive BAL finding as diagnostic of disease? Pl. clearly define the disease entities that you used for categorization in the methods section.

Thank you. We have clarified how we define cryptococcal infection in the methods section.

Figure 2: A denominator is needed in order to be able to make any epidemiological assessment (f.e. hospital days, inpatient admissions).

Thank you. We presented Figure 2 in the format of cases/100 000 admissions/year for an accurate epidemiological assessment as suggested. 

Results: What was the median (range) of follow-up of the cohort, and at what time point did you assess responses to therapy? Pl. clarify.

We included follow-up time of each patient in Table 1. We also reported median and range as suggested in addition to mean follow-up time we reported previously.

Reviewer 3 Report

In this communication the authors examined whether cryptococcal infections in Canada are a result of an epidemiological shift. This was based on other studies that found certain underlying conditions like diabetes, solid organ transplant or hematological malignancies were associated with cryptococcal infections in non-HIV patients. Historically the highest rates of cryptococcal disease has been in patients that were HIV-positive, however, the rates of cryptococcal infections in HIV-patients have dropped due to antiretroviral therapy and improvements in healthcare. Based on this, the authors tested the hypothesis that cryptococcal infections are more prevalent in non-HIV than in the HIV populations. The authors conducted a retrospective case analysis over 12 years of patients who had cryptococcal infections reported at the Ottawa Hospital from 2005 to 2017. The results presented represent 28 patients with cryptococcosis. Based on their findings, the authors conclude that certain underlying conditions in non-HIV patients is a risk factor for the development of cryptococcal infection and further conclude that this suggests a shift in the epidemiology in cryptococcal infections in the Ottawa Hospital.

            Although this is an interesting study and the conclusions are supported by the results, the study reflects only one institution in Ottawa and that is a weakness of the study. Whether these conclusions reflect a similar trend in the city of Ottawa as whole, cannot be deduced by this study alone. Nevertheless, the conclusions discussed in this study are relevant to this particular institute and they do contribute to the overall notion that certain underlying conditions can make patients more susceptible to cryptococcal infections and thus they should be regarded as high-risk and treated preemptively. Some concerns listed below need to be addressed:

Concerns:

Line 132: The authors list the “odds of mortality” – this is a confusing concept given how it is presented, especially since it is not clear how this was calculated and why p values greater than 0.05 would be meaningful in this context.

2. Table 1 – The age, site of positive culture and underlying medical disease are shown in Table 1, however the sex of the patients are not indicated. It would be relevant to the study to include whether the subjects in the study were male or female since HIV and other diseases can be more prevalent in one sex versus another.

Why was this institution in particular examined over others in the city of Ottawa? Is it based solely on the affiliation of the researchers with the institution, or does this Hospital see/treat more cryptococcal disease than other institutions?

The authors should include a conclusions paragraph where they highlight the major findings and underscore the practical aspects for Hospitals and other Institutions in, first identifying non-HIV patients that are at risk of developing a potential fatal cryptococcal disease and then treating these patients preemptively to reduce their odds of acquiring infection.

Author Response

In this communication the authors examined whether cryptococcal infections in Canada are a result of an epidemiological shift. This was based on other studies that found certain underlying conditions like diabetes, solid organ transplant or hematological malignancies were associated with cryptococcal infections in non-HIV patients. Historically the highest rates of cryptococcal disease has been in patients that were HIV-positive, however, the rates of cryptococcal infections in HIV-patients have dropped due to antiretroviral therapy and improvements in healthcare. Based on this, the authors tested the hypothesis that cryptococcal infections are more prevalent in non-HIV than in the HIV populations. The authors conducted a retrospective case analysis over 12 years of patients who had cryptococcal infections reported at the Ottawa Hospital from 2005 to 2017. The results presented represent 28 patients with cryptococcosis. Based on their findings, the authors conclude that certain underlying conditions in non-HIV patients is a risk factor for the development of cryptococcal infection and further conclude that this suggests a shift in the epidemiology in cryptococcal infections in the Ottawa Hospital.

Although this is an interesting study and the conclusions are supported by the results, the study reflects only one institution in Ottawa and that is a weakness of the study. Whether these conclusions reflect a similar trend in the city of Ottawa as whole, cannot be deduced by this study alone. Nevertheless, the conclusions discussed in this study are relevant to this particular institute and they do contribute to the overall notion that certain underlying conditions can make patients more susceptible to cryptococcal infections and thus they should be regarded as high-risk and treated preemptively. Some concerns listed below need to be addressed:

Thank you. Data from a single center is a limitation for generalisability. However, the Ottawa Hospital is a referral center for the region and provides a complex care as well as transplantation (hematopoietic stem cell, kidney and heart). Patient demographic is similar to other tertiary hospitals in Canada. It would be informative to look at the data from multiple centers. Nevertheless, this study is served as an objective evidence to support clinicians’ perception of the changes in epidemiology

Concerns:

Line 132: The authors list the “odds of mortality” – this is a confusing concept given how it is presented, especially since it is not clear how this was calculated and why p values greater than 0.05 would be meaningful in this context.

 Thank you. We thought that presenting an odd of mortality would provide the readers a sense of how much risk of death a patient with non-HIV who developed cryptococcosis has as compared to an HIV-infected patient. We provided 95% confidence interval and p-value to assist the readers of its significance given the small sample size. However, we agree that this information may confuse the readers. We therefore removed the report on odds of mortality.

Table 1 – The age, site of positive culture and underlying medical disease are shown in Table 1, however the sex of the patients are not indicated. It would be relevant to the study to include whether the subjects in the study were male or female since HIV and other diseases can be more prevalent in one sex versus another.

Certainly. We included an additional column of sex in the Table.

Why was this institution in particular examined over others in the city of Ottawa? Is it based solely on the affiliation of the researchers with the institution, or does this Hospital see/treat more cryptococcal disease than other institutions?

Thank you. We included the explanation why data from the Ottawa Hospital was chosen in the Methods. The Ottawa Hospital is a tertiary care center and a referral center for Ottawa city, Eastern Ontario and parts of Northern Ontario. Complex care and treatments such as chemotherapy and transplantation in this region occur at the Ottawa Hospital.

The authors should include a conclusions paragraph where they highlight the major findings and underscore the practical aspects for Hospitals and other Institutions in, first identifying non-HIV patients that are at risk of developing a potential fatal cryptococcal disease and then treating these patients preemptively to reduce their odds of acquiring infection.

Thank you for your suggestion. We included these points in the conclusion paragraph.

Round 2

Reviewer 1 Report

the authors have sufficiently answered my concerns and have provided the necessary information that I desired. The discussion is much improved and answered my concerns. I believe the manuscript is acceptable for publication with their changes.

Reviewer 2 Report

The authors have made the appropriate modifications - I have no further queries.